# Fasting Finisher Pigs before Slaughter Influences Pork Safety, Pork Quality and Animal Welfare

**DOI:** 10.3390/ani10122206

**Published:** 2020-11-25

**Authors:** Bert Driessen, Louis Freson, Johan Buyse

**Affiliations:** 1Research, Control & Consult-Belgium bv, B-3583 Paal, Belgium; bert.driessen@kuleuven.be; 2Laboratory of Livestock Physiology, Department of Biosystems, B-3001 KU Leuven, Heverlee, Belgium; 3Agricultural Buildings Research, Department of Biosystems, B-3001 KU Leuven, Heverlee, Belgium; louis.freson@kuleuven.be; 4Laboratory of Humane Biology, Department of Physiology, B-3000 KU Leuven, Leuven, Belgium

**Keywords:** animal welfare, fasting, pigs, slaughter

## Abstract

**Simple Summary:**

Fasting prior to slaughter is an accepted best management practice in pork production worldwide. Often, fasting time is confounded with the resting time at the slaughterhouse. In practice, fasting is sometimes only applied from the departure from the farm to the slaughterhouse. When implemented correctly with attention to the local factors, pre-slaughter fasting can improve animal welfare, pathogen risk and carcass hygiene. This review aims to describe the impact of fasting pigs before slaughter.

**Abstract:**

The final phase in pork production is the transport of finisher pigs to the slaughterhouse. Fasting is one of the parameters that influence the stress coping ability of the pigs during transport and lairage. When implemented correctly with attention to the local factors, pre-slaughter fasting can improve animal welfare, pathogen risk and carcass hygiene. The length of pre-slaughter feed withdrawal time is important to the success of the production practice. In practice, a fasting time before slaughter between 12 and 18 h enhances pork safety, pork quality, and animal welfare. This means that communication between producer and slaughterhouse is essential when planning the fasting and lairage times to avoid carcass and technological pork quality problems (such as pale, soft, and exudative (PSE) meat or dark, firm and dry (DFD) meat).

## 1. Introduction

The final phase in pork production is the transport of finisher pigs to the slaughterhouse. From the animal point of view, it is a very complex and stressful event. Animal stressors vary as a function of, for example, housing, climate, and handling technique, and have an impact on the animal. Fasting is one of the parameters that influence the stress coping ability of the pigs during transport and lairage (Figure 1). The usefulness and impact of fasting differ from species to species. For example, the gastrointestinal anatomy, fatigue, and stress susceptibility influence the need for and the impact of fasting [1]. Therefore fasting guidelines are made species-specific.

Fasting is a common practice for the on-farm preparation of pigs before slaughter. During fasting, pigs are denied access to feed before slaughter. However, it must be pointed out that fasting does not mean that drinking water is excluded. On the contrary, drinking water should be available to the pigs whenever possible in the fasting period (in particular at the farm and in lairage).

The metabolic response to the challenge of feed withdrawal can be split into two phases. The first phase is typified by the use of circulating glucose and liver and muscle glycogen stores, as well as the decrease in insulin secretion, which affects the meat quality [2,3,4]. Glycogenesis and lipolysis are stimulated in the second phase. There is an increase in fat mobilization and the organism goes into a state of normoglycemia and activates mechanisms that have an impact on the hypothalamus, the nerve endings, the adrenal glands, and the pancreas [5,6].

Fasting time is often confounded with the resting time at the slaughterhouse. In practice, fasting is sometimes only applied from the departure from the pig farm to the slaughterhouse [7,8]. The reason for this misapplication is the producers’ concern about the risk of the reduced growth rate of pigs remaining in the pen and the lack of a holding pen where the heaviest pigs can be transferred until the moment of loading and can be fasted at the same time [8]. Pigs intended for slaughter should be identified and segregated from pen mates with sufficient time to apply the appropriate feed withdrawn to the slaughter group. This review aims to describe the impact of fasting pigs before slaughter (Table 1).

## 2. Effects of Fasting

### 2.1. Weight Loss

In the first 24 h of fasting, pigs can lose up to 5% of their live weight, at an approximate rate of 0.25 kg per hour [35]. In that period live weight losses are more related to the excretion of urine and faeces rather than to body tissues and consequently carcass weight is not affected [36]. This can be clarified by the fact that feed consumed within 9–10 h before slaughter remains in the gut at the point of evisceration and is not converted into carcass weight gain [37].

Live animal weights are increased by feeding the pigs before slaughter, but it does not influence carcass weight. Between four to eight hours after intake by the animals, feed will be absorbed in the small intestine, and most of the nutrients will be transported into the bloodstream nine hours later [38]. Therefore, feed consumed by pigs just hours before slaughter may be considered to be wasted. Consequently, fasting the animals on the farm before transport contributes to a saving of 2 kg of feed/pig for the farmer [39]. Extended lairage time decreases carcass yield because of the impact on the fasting time [14]. If pigs miss one or more meals in a 24 h period, they do not compensate for the missed feed intake. When the fasting is longer than 24 h, it induces catabolism of body stores in finisher pigs.

### 2.2. Gastro-Intestinal Tract

The amount of waste to be disposed of at the slaughterhouse can be decreased by a withdrawal period of up to 48 h [40]. Lower stomach weights have been determined in pigs fed ad libitum compared to pigs with restricted feeding [41,42]. This difference in stomach weights is caused by the intake of smaller feed portions at each meal in ad libitum fed pigs, which favours feed digestion and consequently accelerates stomach emptying [43].

According to Guise et al. [44], the gastric emptying rate in fasted pigs is affected by the smaller feed particle size and pelleting. Feed composition can also affect the efficiency rate, as shown by Magras et al. [41] in pigs fed corn or wheat-based diets in 3 to 4 meals/day and fasted for 22 h before slaughter. Wheat-based diets slow down gastric emptying due to their higher fibre and carbohydrate content. Wheat-based diets show a lower efficiency rate value (87.9%) compared to maize-based diets (95.4%). Thus, a withdrawal period longer than 22 h is advised for pigs fed wheat-based diets to increase the efficiency of fasting before slaughter and to minimise intestinal fill and the risk of intestinal perforation during the slaughter and evisceration process. Thus, not only the stomach weight must be monitored in audits for potential sources of carcass contamination at the slaughterhouse, but also the stomach content type [45].

Pigs consume the majority of the daily feed intake in the 12 h between approximately 0600 h and 1800 h [46,47]. This affects the timing of the beginning of fasting and its duration. Pigs that start fasting at 0600 h compared to 1800 h probably have vastly different amounts of feed in the stomach and gastrointestinal tract.

### 2.3. Contamination

Food safety is the primary consumer issue and the overriding issue for public responsibility in the marketing of meat. Feed withdrawal accounts for ±70% of the variation in the carcass contamination rate [9]. The greater the content of the gastro-intestinal tract at slaughter, the higher the risk of lacerating these tissues during evisceration, and the higher the risk of carcass contamination. The shedding rate of Salmonella in animals is known to be increased with both feed withdrawal time and stress [10,11]. The presence of manure on the truck or lairage pen floor is an important potential source of faecal bacteria in carcass contamination. Stressors promote the rate of gastric emptying, the proliferation of *E. coli*, and the Salmonella population in the intestines and their excretion into the environment [48,49,50,51]. According to Berends et al. [52], carcasses of live pigs carrying Salmonella at the time of slaughter are 3 to 4 times more likely to test positive for Salmonella than carcasses of Salmonella-free animals.

### 2.4. Behaviour

Some reports [42,53] indicate that pigs subjected to pre-slaughter feed withdrawal are easier to move and handle during loading, transporting, and unloading. The results of Dalla Costa et al. [17] are in contrast with these findings. Dalla Costa et al. [17] studied the impact of 24 h fasting before slaughter in two groups of pigs with different fasting regimes. One group fasted 18 h at the farm before transport and 6 h before slaughter (including 2 h transport and 4 h lairage), while the other group started fasting at the departure from the farm (2 h transport plus 22 h lairage). The pigs fasted at the farm showed more going backwards and 180° turn behaviours, and vocalized more during loading. In addition, Acevedo-Giraldo et al. [22] described in their research that pigs fasted for 8 h on the farm when driven, balked or attempted to turn around more frequently compared with non-fasted pigs. Hunger and feed restriction induces frustration and excitement and explains the balking behaviour [17].

According to Dalla Costa et al. [17], pigs fasted at the farm tended to have a shorter latency to lie down in the lairage of the slaughterhouse. During the first 2 h of lairage, a greater proportion of farm fasted pigs were lying, while pigs fasted at the slaughterhouse showed a higher number of fights and a longer total duration of fights for several periods. Acevedo-Giraldo et al. [22] noticed that pigs without feed restriction had a greater proportion of agonist behaviour during the lairage period. The shorter latency to lie down of farm fasted pigs in the lairage may suggest their need to recover from physical exhaustion caused by the additive effect of feed restriction, handling, and transportation [54]. In this metabolic condition, pigs would rather rest than fighting with the new pen-mates or exploring the pen. Brown, Knowles, Edwards, and Warriss reported contrasting results [55]. They noticed a higher fighting and drinking activity on arrival at the lairage pen in pigs fasted for a longer time (18 h vs. 1 h fasting). The contradiction of the impact of the feed withdrawal time can possibly be explained by the confounding interaction between the feed withdrawal time and the resting time in lairage.

### 2.5. Skin Damage

Longer lairage increases the incidence of skin damage because pigs were more aggressive due to the prolonged period of feed withdrawal [18,19,20,21]. In this case, a longer lairage is correlated with a longer feed withdrawal time. However, Dalla Costa et al. [17] also reported a pure fasting effect. They showed that fasted (18 h) pigs displayed a shorter total duration of fights and a smaller proportion of fights during lairage than non-fasted pigs. The non-fasted pigs continued to fight with the same intensity up to the third hour of lairage. It can be concluded that fasting pigs at the farm influences the welfare of the pigs later on in the slaughterhouse.

### 2.6. Meat Quality

Pre-slaughter fasting can affect meat quality by an increase of pHu [56,57,58]. Conversely, several other studies [36,37,59] described no or limited impact of feed withdrawal time on meat quality. The differences in the stress level or the activity imposed on pigs before slaughter and by the muscle being used for the meat quality assessment in each study may explain the discrepancy among the reported results [38]. When fasting is not confounded by other pre-slaughter practices (e.g., mixing, high ambient temperature, etc.) muscle glycogen does not become depleted to an extent that pork quality is influenced [58,60]. However, the reduction of the muscle glycogen can suffice to affect ultimate pH in the muscles that are involved in the animal’s posture and weight and/or having a lower glycolytic potential [12,61]. Pre-slaughter fasting can be considered a tool for increasing ultimate muscle pH and lowering the incidence of pale, soft, and exudative (PSE) pork [12]. A fasting period of less than 18 h increases the prevalence of PSE meat [13]. On the other hand, long fasting periods (>22 h) induces muscle glycogen exhaustion and raises the risk of DFD (dark, firm, dry) meat [14,15,16].

### 2.7. Mortality

Fasting pigs before slaughter decreases the risk of mortality during transport to the slaughterhouse. Non-fasted pigs demonstrate a higher risk of transport sickness and vomiting during transport than fasted pigs. This situation leads to higher mortality, especially in summer and in animals genetically susceptible to stress [23]. The mortality of unfasted pigs during transport is caused by the pressure of the full stomach on the vena cava, resulting in decreased blood flow efficiency [62]. The mortality rate of transported pigs that have been fasted 8–18 h before loading is significantly lower than that of pigs loaded on a full stomach [23,24]. According to Averós et al. [25], the mortality risk of non-fasted pigs increases with transport up to 8 h. In fasted pigs, Averós et al. [25] noticed no effect of transport duration (up to 24 h) on the mortality risk.

### 2.8. Motion Sickness

Motion sickness has been described in several species, including pigs [63,64]. Pigs show motion sickness-related signs like foaming at the mouth, chomping, retching, and vomiting [26,63,65]. Fasting decreases motion sickness [26,27] and the circulating levels of vasopressin during transport compared to non-fasted pigs [28]. The scientific evidence for an optimal fasting time regarding motion sickness is sparse. The last feed before transporting should be between 4 and 12 h, according to Warriss and Brown [24]. However, in a later publication, Warriss et al. [18] mentioned that four hours of feed withdrawal is too short for trips on poor roads or in vehicles with poor vibration characteristics.

### 2.9. Gastric Ulcers

Eisemann et al. [29] and Swaby and Gregory [30] noticed an effect of feed withdrawal before slaughter on the prevalence of gastric ulcers in pigs. Long fasted pigs (held overnight in the lairage) have a higher risk of severe gastric ulcers than pigs slaughtered on the day of arrival at the slaughterhouse. Stress affects the incidence of gastric ulcers, especially when there is either impaired blood flow to the mucosa or the feeding regime increases the exposure of the mucosa of the pars oesophagea to acidic conditions [66]. The reason for the relationship between a brief period of starvation and gastric ulceration is not known, but it has been hypothesized that without continuous feed intake, the normal pH gradient of the stomach is disturbed because of increased fluidity and mixing of gastric contents. In addtion, there is a reduced pH associated with the absence of feed buffers in the empty stomach [67,68].

### 2.10. Thermal Stress

Under warm summer conditions, especially in temperate climates, pigs are exposed to high temperatures during housing, but also during transport and lairage [69]. Under these ambient conditions, the maintenance of body temperature is achieved by increasing heat loss and reducing heat production [31]. Many factors including pig’s size, duration of fasting, floor type, air velocity, and group size influence the thermoneutral zone for pigs during transport and lairage [32]. According to Koong et al. [33] and van Milgen et al. [34], heat production is generally explained by an indirect effect of reduced feed intake. In conclusion, fasted pigs cope better with thermal stress.

## 3. Discussion

Based on the current data, the fasting of finisher pigs should start on the pig farm and should be applied to reduce diminishing effects on pig welfare, food safety, and meat quality [9,12,23,26,27,29,30,33]. On the other hand, extended fasting times induces carcass yield losses and aggressiveness [14,17,18,19,22]. Thus, in practice, the fasting period is limited by a minimum and a maximum limit. Despite the importance of a fasting period, the guidelines for an optimal fasting period are vague. In the literature, a fasting period of 12 to 18 h is suggested. Sometimes a period of up to 24 h is suggested. CATGP [70] recommends a fasting period of 10 to 12 h before loading in case of transport less than 8 h long. Based on the current literature, it is not clear whether these minimum and maximum limits are sufficiently delimited or should be more limited in the function of specific conditions, for example, in warm summer conditions.

To estimate the total fasting time, time without feed on-farm, loading time, transport time, unloading time, and time in lairage must be taken into account. To determine the start of fasting, one should start from the estimated moment of slaughter and then recalculate to determine the start of the fasting period. In practice, the slaughterhouse determines the moment of slaughter and thus indirectly the start of the fasting in the pig farm. However, the effective start of fasting at the predetermined start moment is no guarantee that the predetermined fasting time will be achieved. For example, a traffic jam or a breakdown in the slaughterhouse can disrupt slaughter planning and consequently the total fasting time [69].

The starting moment of fasting should also be given attention. When feeding ad libitum in feeding troughs, the troughs are closed at the moment, but there is usually feed in the trough for about 1 or 2 h. Of course, the period of fasting starts when the trough is empty. Another possibility is that after closing the trough, the remaining feed is scooped out of the trough so that the period of fasting can be started immediately.

Finisher pigs can be kept in a fully-slatted or a straw-bedded system. In contrast to the pigs in a fully-slatted system, pigs in a straw-bedded system always have access to straw. Pigs kept in enriched housing conditions need a more extended feed withdrawal time to reduce the high glycogen stores to the level required at slaughter to produce normal meat quality, as suggested by Rocha et al. [71]. This hypothesis could not be validated in the study of Faucitano et al. [72]. In their study, the meat quality of pigs fasted for 24 and 32 h were compared. They concluded that the practice of an extended feed withdrawal period up to 32 h pre-slaughter may nullify the advantages of enriched housing conditions and thus cannot be recommended.

In practice, fasting is sometimes only applied from the departure from the pig farm to the slaughterhouse [7,8]. The risk of the reduced growth rate of pigs remaining in the pen and the lack of a holding pen where the heaviest pigs can be transferred to ahead of loading time and feed be withdrawn at the same time induce misapplication by the producers [8]. In response to this producers’ concern, research and production focus should be more on limiting weight differences during rearing to transport pig batches, uniform in weight, to the slaughterhouse [73] or by moving slaughter weight pigs to a separate holding pen.

To our knowledge, very little scientific literature is available that addresses the transportation and slaughter of cull sows and boars. However, pigs at other stages of production probably need specific considerations during transport, due to their physiological conditions or age [74]. It is well known that there is a correlation between the conformation (i.e., muscled) and body condition of pigs and meat quality [75]. More research about the need and impact of fasting on cull sows and boars is needed.

## 4. Conclusions

When implemented correctly with attention to the local factors, pre-slaughter fasting can improve animal welfare, pathogen risk reduction and carcass hygiene. The length of pre-slaughter feed withdrawal time is important to the success of the production practice. Therefore, in practice, a fasting time before slaughter between 12 and 18 h enhances pork safety, pork quality, and animal welfare. This means that communication between producer and slaughterhouse is essential when planning the fasting and lairage times to avoid carcass and technological pork quality problems (such as PSE or DFD).

## Figures and Tables

**Figure 1 animals-10-02206-f001:**
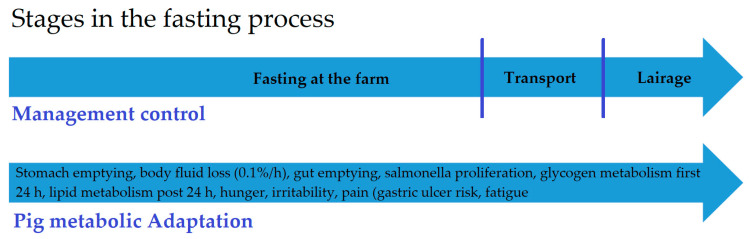
An overview of the different stages in the fasting process of finisher pigs. Because of the good distribution of pig slaughterhouses in Belgium, the mean transport time is 2 h. After transport, pigs are kept in lairage for 2 or 3 h before slaughter. If a total fasting period of 18 h is envisaged, the pigs should fast for 13 h on the farm. In addition, the metabolic adaptions in the fasting process of pigs are given.

**Table 1 animals-10-02206-t001:** Intended and unintended outcomes of pre-slaughter feed withdrawal, the cells were only filled in if the information was relevant.

Outcome	Intended	Unintended	Reference Number
Pork safety	Decrease carcass contamination		[9]
		Increase Salmonella carriage	[10,11]
Pork quality	Optimizing pHu		[12]
		Increase PSE in situations of too short fasting	[13]
		Increase DFD in situations of too long fasting	[14,15,16]
Animal Welfare	Decrease skin damage		[17]
		Increase skin damage	[18,19,20,21]
	Decrease fighting		[17,22]
	Decrease dead in truck		[23,24,25]
	Decrease nausea		[26,27,28]
		Pain gastric ulcer	[29,30]
	High thermal stress tolerance		[31,32,33,34]

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
