# Peer review of "Fasting Finisher Pigs before Slaughter Influences Pork Safety, Pork Quality and Animal Welfare"

_animals, 2020, doi:10.3390/ani10122206_

Round 1

Reviewer 1 Report

Line 14-15 Clarify/reword “In practice, fasting is sometimes only applied from the departure from the farm to the slaughterhouse in pig farms.” Suggest deleting “in pig farms”

Line 25 Clarify “technological pork quality problems”

Line 33-34 “The usefulness and impact of fasting differ from species to species. For example, the 33 gastrointestinal anatomy, fatigue, and stress susceptibility influence the need and the impact of 34 fasting.” Reference?

Line 49 “shipping room” more commonly referred to as “holding pen”

Line 71 “2 kg of feed/pork” should read “2 kg of feed/pig”

Line 72 “Extended lairage decreases carcass” insert “time” between “lairage” and “decreases” i.e Extended lairage time decreases carcass . . .

Line 97 Reword “Food safety is since decades a defining issue in the meat market.”

Line 98-100 “The greater the content of the gastro-intestinal tract at slaughter, the higher the risk of lacerations during evisceration, and the higher the risk of carcass contamination.” Replace “lacerations” with “lacerating these tissues”

Line 116-117 “Also, Acevedo-Giraldo et al. [32] described in their research that pigs 8 h fasted on the farm backwards more frequently compared with non-fasted pigs.” Should read “Also, Acevedo-Giraldo et al. [32] described in their research that pigs fasted for 8 hr on the farm moved backwards more frequently compared with non-fasted pigs.”

Line 130 ”withdrawal time can maybe be explained by the . . “  Replace “maybe” with “possibly”

Line 139-140 “It can be concluded that fasting pigs at the farm influences the welfare of 139 the pigs later on in the slaughterhouse.” Your conclusion is based on the assumption that fasting per se is less stressful that fighting, have you any evidence for this?

Line 196 “In conclusion, fasted pigs coop better with thermal stress.” Replace “coop” with “cope”

Line 200 “too long fasting induces carcass” insert “times” between “fasting” and “inducing”

Line 212 “indirect the start of” replace “indirect” with “indirectly”

Line 234-235 “differences during rearing to transport pig batches, uniform in weight, to the slaughterhouse 234 [72].” Insert “or, by moving slaughter weight pigs to a separate holding pen.” After “[72]”

Line 249 “technological pork quality problems” see comment on Line 25

Reviewer 2 Report

See attached file
